# Unwanted Scratching Behavior in Cats: Influence of Management Strategies and Cat and Owner Characteristics

**DOI:** 10.3390/ani12192551

**Published:** 2022-09-24

**Authors:** Alissa Cisneros, Dorothy Litwin, Lee Niel, Anastasia C. Stellato

**Affiliations:** 1Department of Animal and Food Sciences, Texas Tech University, 2500 Broadway, Lubbock, TX 79409, USA; 2Department of Population Medicine, University of Guelph, 50 Stone Rd E, Guelph, ON N1G2W1, Canada

**Keywords:** cat, scratching, enrichment, training, welfare, behavior

## Abstract

**Simple Summary:**

Cat scratching is a self-motivated and natural behavior, yet cat owners commonly report unwanted scratching on household items, such as furniture, walls, and carpets. This study explored the influence of owner management strategies, owner characteristics, and cat characteristics on the performance of unwanted scratching in the home. Perspectives of various intervention and management methods were collected using an online survey (*n* = 2465). Results reveal that owners generally agreed with providing additional appropriate surfaces and items for their cat to scratch rather than more permanent solutions, such as surrendering, euthanizing, or declawing. There were fewer reports of unwanted scratching for cat owners with older cats, and for those that provide enrichment items (e.g., sisal rope), outdoor access, positive reinforcement training, and restrict access to items they did not want scratched. Through this study, results suggest that cat owner interventions and management strategies can influence reports of unwanted scratching in cats and encourages the continued education to the public on effective options available.

**Abstract:**

Despite scratching behavior in owned domestic cats being a self-motivated and natural behavior, it is commonly reported as a behavior problem by owners when it results in damage to household items. The objectives of this study were to use a cross-sectional survey targeting cat owners within the United States and Canada, to explore perspectives on cat scratching behavior and management strategies, as well as identify factors that influence the performance of inappropriate scratching behavior in the household. A total of 2465 cat owners participated in the survey and three mixed logistic regression models were generated to explore associations between (1) cat demographic factors, (2) provisions of enrichment, and (3) owner demographic and management factors with owner reports of problematic scratching. In this convenience sample, inappropriate scratching was reported by 58% of cat owners. Owner perspectives and management strategies aligned with current recommendations as they preferred to use appropriate surfaces (e.g., cat trees) and training to manage scratching as opposed to surrendering, euthanizing, or declawing. Logistic regression results found fewer reports of unwanted scratching behavior if owners provide enrichment (flat scratching surfaces (*p* = 0.037), sisal rope (*p* < 0.0001), and outdoor access (*p* = 0.01)), reward the use of appropriate scratching objects (*p* = 0.007), apply attractant to preferred items (*p* < 0.0001), restrict access to unwanted items (*p* < 0.0001), provide additional scratching posts (*p* < 0.0001), and if their cat is 7 years of age or older (*p* < 0.00001). Whereas if owners use verbal (*p* < 0.0001) or physical correction (*p* = 0.007) there were higher reports of unwanted scratching. Results suggest that damage to household items from scratching behavior is related to management strategies owners employ, and these findings can be used to support owner education in mitigation and prevention of inappropriate scratching.

## 1. Introduction

Cats have become one of the most popular household pets, as recent estimates suggest there are approximately 62 million owned cats in the United States (US) and 8.1 million in Canada [1,2]. Cat behavior issues can be problematic for both cat welfare and the cat-owner relationship. Scratching is one of the most reported issues with the proportion of owners reporting scratching as a behavioral issue ranging from 5.4% [3]–15.2% [4], to 30% [5,6]–84% [7]. Inappropriate scratching is likely perceived as a problem for some cat owners as it can cause damage to household items, such as chairs, furniture, carpet, sofas, doorways, walls, and curtains [7].

Scratching is a highly motivated, natural behavior with various functions, such as marking territory for communication with other cats through pheromones found in the interdigital glands, sharpening claws, and removing claw sheaths (the outer layer of the claw) [8]. Given that cats need an outlet for scratching, it is important to understand owner perceptions and management strategies of scratching behavior to avoid undesirable outcomes for both cat and owner. A variety of management solutions have been suggested to control scratching behavior, such as frequent nail trims, applying nail caps, and providing enrichment items [9,10]. Alternatively, owners may seek a more permanent surgical solution, such as declawing [11]. According to data collected from a US veterinary hospital, approximately 21% of cats are declawed [12]. In one online survey examining the influence of external factors (such as the environment, scratching posts provided, and cat demographics) on inappropriate scratching, 34.5% of 386 owners of declawed cats reported they elected to declaw their cats to prevent damage to household items [13]. Other consequences of prolonged scratching issues in homes include euthanasia and shelter surrender [14,15,16], with a previous study indicating that scratching behavior problems are associated with an increased risk of relinquishment, and declawing cats decreases the risk of relinquishment to shelters [16].

Cats in the home have opportunities to scratch a variety of surfaces including those provided for enrichment and off-limit items, such as furniture. Research suggests that cats have preferences toward different substrates, include chenille fabric [17], cardboard, rope, and carpet [13,18,19]. Provisioning of scratching materials (e.g., rope, cardboard, carpet, wood surfaces, vertical and flat/horizontal scratching posts) to cats is widely recommended to encourage appropriate scratching and reduce damage to household items [13,19,20,21]. In addition, placing the cat near an appropriate scratching surface has been found to promote appropriate scratching [7,21], while punishment-based methods, such as physical and verbal scratching interruption, has not been effective [7,13]. To encourage scratching on appropriate items, current recommendations include applying catnip and specific pheromones (e.g., Feliway) on scratching objects and materials [19]. The application of feline pheromones (specifically, feline interdigital semiochemical marketed as Feliscratch by Feliway) has been found effective in reducing unwanted scratching [22], with one study reporting 74% of cats redirecting their scratching toward the treated scratching posts and away from household items [23]. The utilization of positive reinforcement to encourage the use of these items has anecdotally shown promise in reducing the time it takes for cats to interact with new objects [24]. It is believed that inappropriate scratching has become more problematic within households as a result of restricting cats indoors [8,10]. Provision of outdoor access, whether controlled (e.g., in a controlled area or under direct supervision) or uncontrolled (free roaming and unsupervised), is believed to potentially help to reduce unwanted scratching with its additional opportunities and items available for scratching, such as trees, posts, and fences [25,26]. Certain cat characteristics influence the incidence of inappropriate scratching; for instance, inappropriate scratching is known to decrease with age and reports suggest no difference in scratching behavior between intact or neutered male and female cats [13]. Regarding owner characteristics, it is unknown as to how the owner influences scratching behavior beyond the environment and cat care provided.

The aim of this study was to identify factors that influence the performance of inappropriate scratching in owned cats and to understand owner perspectives on managing scratching issues in the home. Using predictors based on current literature, an online survey was developed regarding characteristics of scratching behavior, cat and owner-related demographics, and associated intervention methods employed by owners.

## 2. Materials and Methods

### 2.1. Data Collection

To be eligible to participate, individuals were required to be 18 years of age or older, the primary caregiver of at least one cat (e.g., financially responsible), and a resident of Canada or the US. The survey was created and made available to participants through Qualtrics^®^ and distributed online from July to October 2019. Virtual snowball sampling was used to distribute the survey, as the initial advertisement (which called owners to participate in a survey regarding scratching) was shared via Facebook and sharing was encouraged. To encourage outreach, the advertisement was shared to other behavior and welfare and cat-related social media pages. This referral-based method of recruitment has been demonstrated to be effective in reaching certain groups that may otherwise not be captured [27,28].

### 2.2. Questionnaire

The questionnaire consisted of 43 questions divided into five sections: cat characteristics (e.g., sex, age, breed, acquisition source, neuter status); home environment (e.g., time spent playing with cat, time spent interacting other than playing, provision of outdoor access); owner management of and perspectives on scratching behavior (e.g., provision of scratching materials/objects, cat preferences for scratching materials/objects, owner perspectives about scratching, and strategies used to mitigate scratching); declaw status (e.g., cat age when performed, type of procedure, reason why it was performed); owner demographics (e.g., gender, education, age, advanced experience and education related to cat care, number of cats owned). Participants who owned more than one cat were asked to respond to the survey for the cat whose name begins with the letter closest to the beginning of the alphabet for all relevant questions.

### 2.3. Statistical Analysis

All analyses were performed with Stata Statistical Software v.17.0 (StataCorp., College Station, TX, USA).

#### 2.3.1. Data Management

A total of 35 variables were used for analysis, and were related to the following three categories, (1) cat demographic factors (e.g., cat age, breed, neuter status), (2) provisions of enrichment (e.g., type of outdoor access, scratching materials/objects, active time playing), and (3) owner demographic and management factors (e.g., age, gender, experience, response to observed scratching—provision of additional scratching objects, place deterrents to prevent scratching [double-sided tape, tin foil, deterrent spray], restrict access to off-limit scratching objects, verbal correction [yelling ‘No’], physical correction [spray with water, tap or smack], interrupt scratching [redirect to appropriate object, call them or move them away], reward for using appropriate objects, apply an attractant to preferred areas, trim nails, apply nail caps). To reduce the number of variables tested for analysis, related variables were collapsed to create composite variables. For instance, the variable ‘experience’ was created from the following variables: cat sitter, cat breeder, cat trainer, veterinarian, veterinary technician, cat boarder, shelter worker, groomer, researcher, and foster parent. As these variables were all directly linked to one survey question regarding cat-related experience, these variables were highly conceptually related and thus a reliability analysis was not deemed necessary. All independent variables were categorical, with cat age categorized based on the cat life stages presented in the AAFP 2010 guidelines (less than 4 months, 4–12 months, 1–6 years, 7–10 years, 11–14 years, and greater than 15 years [29]), and participant age categorized based on biological/practical cut-points (18–24, 25–44, 45–64, 65+).

#### 2.3.2. Logistic Regression Models

Three logistic regression models were created to evaluate associations between owner reports of problematic scratching and independent variables related to (1) cat demographic factors, (2) provisions of enrichment, and (3) owner demographic and management factors. The outcome variable, problematic scratching, was based on owner reports on whether their cat has caused damage within the past month to off-limits household items through scratching, where ‘off-limits’ was defined as anything they would not want their cat to scratch. Participants were able to select either no damage, minor damage, or moderate/major damage to this question. To complete a logistic regression analysis, the outcome variables were re-categorized to include levels ‘Yes’, which was a combination of any level of damage reported, and ‘No’.

To further reduce the number of variables included in the multivariable models, a univariable analysis was performed to test each independent variable against the outcome, problematic scratching. Variables were retained for analysis using a liberal *p*-value of *p* ≤ 0.20 [30]. Correlation analysis was performed on all retained variables, with a correlation coefficient of greater than |0.7| suggesting collinearity [31]. The final main effects models were created using a stepwise backward selection process, where significant variables (*p* < 0.05) were kept in the final model. Two-way interactions between biologically plausible variables and potential confounding variables were tested. Confounders were identified as a variable that caused greater than 20% change in a coefficient of another variable in the model. No significant interactions or confounders were detected. To account for clustering within province and state, a variable including a list of provinces and states was tested as a random effect. Province/state was not significant when evaluating the influence of owner demographic and management factors on problematic scratching and was therefore removed. Model fit was determined using Bayesian information criterion.

## 3. Results

### 3.1. Descriptives

The questionnaire was answered by 2826 total respondents. A total of 2465 participants remained and was included in analysis after excluding 361 participants that reported to have a declawed cat. Most respondents resided in Canada, making up 88% of the responses, while residents of the US accounted for 12% of responses. Participant mean (SD) age, in years, was 40.9 (14.1) (range: 18–83 years), and a majority of respondents identified as a woman (94.3%), with a minority identifying as a man (4.8%).

Damage to household items due to scratching was reported by 57.5% (1406/2447) of respondents. Unwanted scratching was reported to occur on the following off-limits items: furniture (85%; 1204/1417), carpets (38.7%; 548/1417), walls (12.4%; 176/1417), curtains (10.8%; 153/1417), and other household objects (9.9%; 140/1417). For owners that reported damage due to scratching, 8.1% strongly agreed that they were bothered by it, while 32.3% strongly disagreed (Table 1). Regarding management, most owners strongly disagreed that they have considered euthanizing (99.2%), surrendering (96.5%), or declawing (92.1%) their cat due to their scratching behavior. Additionally, many respondents strongly agreed that scratching could be resolved through provision of appropriate surfaces (36.3%), training (33.5%), frequent nail trims (28.5%), and nail caps (25%). Further information regarding owner perspectives and management methods for unwanted scratching is detailed in Table 1.

Regarding enrichment items provided, 72% of owners who reported unwanted scratching also reported confining their cat indoors compared to only 67% of those without unwanted scratching in the home. The most common enrichment materials owners reported providing included cardboard, sisal rope, and carpet, and the most reported enrichment objects included scratching posts, cat trees, and flat scratching surfaces (flat surface that lies in a horizontal position on the ground). For a full description of enrichment items provided, see Table 2.

### 3.2. Risk Factors

Risk factors for unwanted scratching are presented in Table 3. The final models included provision of enrichment (sisal rope, flat scratching surfaces, outdoor access), owner management methods (techniques used to discourage unwanted scratching and encourage appropriate scratching), and cat demographic information (cat age). Random effects for province/state were significant but numerically small for both enrichment and cat characteristic models.

#### 3.2.1. Enrichment Factors

If owners reported their cat use fabric materials (e.g., cotton) for scratching, owners were more likely to report unwanted scratching behavior compared to those without unwanted scratching behavior. Fewer reports of unwanted scratching were associated with provisioning of outdoor access and the use of sisal ropes and flat surface scratching materials (Table 3).

#### 3.2.2. Owner Demographic and Management Factors

Owners were less likely to report unwanted scratching if they provided additional scratching posts, restricted access to unwanted scratching items, rewarded the use of appropriate scratching objects, and applied attractant to preferred scratching objects (Table 3). More reports of unwanted scratching were associated with the use of verbal and physical correction and interrupting scratching of unwanted items. No other owner demographic factor (age, gender, and experience with cats) was significantly associated with reports of unwanted scratching.

#### 3.2.3. Cat Characteristics

Cat age was associated with owner reports of unwanted scratching, where cats older than 7 years of age were less likely to have unwanted scratching reported in comparison to cats 1–6 years of age (Table 3). No other cat characteristic (sex, neuter status, source, breed, age acquired, and number of cats in the household) was significantly associated with unwanted scratching.

## 4. Discussion

Regardless of the presence of problematic scratching, most owners from the current study agree that providing enrichment (e.g., cardboard, sisal rope, carpet, scratching posts, cat trees, and flat scratching surfaces) is important to promote appropriate scratching behavior. Although cats may not use all the enrichment items provided, owner provisioning of these items likely reflects owner awareness of potential benefits. A previous study similarly reported 76% of owners have items designated and available for the cat to scratch (e.g., posts, poles, cat trees and towers, cardboard, newspaper, wicker baskets, carpet), further indicating owner awareness of the importance of providing these objects [7]. Despite owner reports of unwanted scratching in the current study, most owners did not report that they were bothered by the scratching, and very few owners reported that they had considered surrendering, euthanizing, or declawing their cats in response to their cat’s scratching behavior. Thus, while scratching is a common behavioral issue for cats, it does not appear to be perceived as a significant problem for owners. However, this finding could potentially be influenced by survival bias, since the sample does not include declawed cats or cats who had already been euthanized for scratching behaviour. A longitudinal study following kittens through adulthood is necessary to fully understand whether some owners find this behaviour problematic.

### 4.1. Enrichment Factors

Logistic regression results suggest that providing outdoor access (via enclosure, on leash, tethered, or roaming free) has a protective effect for unwanted scratching, and this is likely because cats have more opportunities, surfaces, and materials to scratch outdoors compared to when confined indoors. When cats are kept indoors, the current findings suggest that providing enrichment may serve to reduce problematic scratching. This is similar to previous research that showed the positive influence of exposing cats to a variety of enrichment items in managing other common behavior problems, such as aggression and inappropriate elimination [32]. The only scratching materials associated with reports of unwanted scratching were sisal rope, flat surfaces, and fabric materials, with flat surfaces and sisal rope associated with fewer reports and fabric material associated with more reports of unwanted scratching. While the use of fabric material was associated with more reports of unwanted scratching, it may be provided to cats in response to displays of scratching and therefore used at higher rates by cats who already perform unwanted scratching. Similarly, in a recent study exploring cat preferences for fabrics commonly used to upholster furniture, a preference for chenille fabric was detected [17]. As no other type of scratching material was associated with unwanted scratching, these materials could provide an indication of preferred materials for scratching. Each type of scratching material was tested on their own, as such the overall effect of providing scratching material was not examined as provisioning of general enrichment does not necessitate their use; further, examining the use of each type of enrichment item was determined to be more informative regarding the influence of each item on scratching behavior. Cats use scratching for a variety of reasons, including communicating with other cats for territory marking [8], thus further research may explore whether the type and location of scratching material influences their scratching behavior. Additional research is needed to understand the relationship between feline preferences for scratching material and the purpose for this behavior, as well as how it might impact scratching behavior in the home.

### 4.2. Owner Demographic and Management Factors

Positive reinforcement practices (rewarding the use of appropriate scratching objects) were associated with reduced unwanted scratching and the use of positive punishment (verbal and physical correction) was associated with increased displays of unwanted scratching. A previous study surveyed cat owners at a single clinic to explore frequency of inappropriate scratching and intervention methods, and found that punishment (such as yelling, using a spray bottle, physical correction, and using loud noises) did not deter scratching from off-limit items, while placing the cat near the desired scratching object decreased scratching-related damage [7]. Similar findings of positive punishment methods being associated with the performance of undesirable behaviors (e.g., aggression) have been observed in both cats [33] and dogs [34,35,36,37,38]. Positive reinforcement training has been determined as the best practice and when used for cats, it has been found to result in a faster reaction time for the desired response [24]. Though results suggest that positive punishment may increase unwanted scratching, it could also suggest that cats that perform unwanted scratching are more likely to receive physical or verbal correction from their owner; further research is needed to determine the direction of this relationship.

Regarding management strategies, the current results suggest that restricting access to inappropriate scratching items, applying attractants (e.g., feline pheromones and catnip) to appropriate scratching items, and providing additional scratching posts is protective against the performance of unwanted scratching. These findings support current recommendations for the use of these simple and inexpensive strategies to prevent unwanted scratching. Similar research has observed that the use of attractants increases the rate of scratching on appropriate household items [19,22,23], and inappropriate scratching reduces as the number of scratching posts increase in the home [13]. Interrupting scratching of household items was a management method associated with increased reports of problematic scratching; however, it may be likely that owners are more likely to interrupt their cats when they are scratching on off-limits household items. The use of deterrents (e.g., double sided tape, tin foil), nail trims, and nail caps were not associated with unwanted scratching, suggesting these strategies do not have an influence on increasing or decreasing owner reports of unwanted scratching behavior.

### 4.3. Cat Characteristics

Owners with cats older than 7 years of age had fewer reports of unwanted scratching, and cats between 4–12 months old had the most reports. These findings might be explained by variability in cat activity levels with developmental stage, as older cats generally have lower activity levels and spend more time sleeping, while kittens are reported to have higher activity levels, spend more time exploring their surroundings, and may not have learned which surfaces are appropriate yet [39,40,41]. Additionally, this finding may be a result of survival bias as older cats may have been removed from our sample as a result of declaw (as declawed cats were removed from our sample), relinquishment, or euthanasia. Further longitudinal research is necessary to explore changes in scratching behavior and owner responses to scratching over time.

### 4.4. Limitations

As observed in most online surveys [42], the majority of respondents were female, reducing the generalizability of results to male cat owners. Respondents also resided largely in Ontario, Canada, limiting the generalizability of the study to American cat owners. Further, the distribution of the survey was conducted through social media that reached other cat-related social media pages and other animal welfare organizations. This audience represents a population that could be more aware of the appropriate behavioral interventions and management strategies needed for cats than the general population. This could influence the rate of agreement with certain methods, such as positive punishment versus positive reinforcement. The results could reflect a bias within a more informed population; however, it unlikely influences model findings as they are not linked to owner demographic variables. Sampling error could also have arisen from the exclusion of participants who reported declawing their cat, though this was deemed necessary as these participants would not be able to report damage to scratching. Additionally, by combining the different levels of cat experience into a single variable, ‘experience’, we were unable to confirm whether different levels of experience influence different perceptions and different management strategies employed by cat owners; for future research a more detailed analysis would be needed to determine this. In addition, due to the way the survey was conducted online, the results were limited to cat owners with internet access. Owner bias could be reflected in the results due to the social desirability bias, as cat owners may have altered their responses to appear to provide greater welfare for their cats. A longitudinal trend could not be captured due to the survey being cross-sectional, thereby preventing the ability determine causation within the study. Further research is therefore needed to identify the effectiveness of various intervention methods on the incidence of inappropriate scratching in the home.

## 5. Conclusions

Unwanted scratching was reported by a majority of respondents, but most owners were not bothered by scratching behavior in the home. Cat owner perspectives aligned with current recommendations regarding the provision of appropriate enrichment and management techniques. Results suggest that owner management strategies for unwanted scratching behavior may influence the incidence of damage to household items; for instance, the use of positive punishment techniques was associated with more reports of unwanted scratching while providing enrichment and positive reinforcement techniques was associated with fewer reports of unwanted scratching. Owner education on the use of positive reinforcement training for behavioral management could benefit those seeking to reduce unwanted scratching in the home. The current study highlights the need for future interventional and longitudinal research to explore the influence of these management techniques on the performance of unwanted scratching behavior in the home.

## Figures and Tables

**Table 1 animals-12-02551-t001:** Summary statistics comparing perspectives and management strategies for resolving unwanted scratching for cat owners who reported that their cat performs unwanted scratching (*n* = 1406) or no unwanted scratching (*n* = 1041).

Variable	Levels	Unwanted Scratching	No Unwanted Scratching
**Perspectives on Managing Scratching**			
Provision of appropriate surfaces	Strongly Agree	503 (36.3%)	692 (67.6%)
	Somewhat Agree	661 (47.7%)	275 (26.9%)
	Somewhat Disagree	134 (9.7%)	15 (1.5%)
	Strongly Disagree	88 (6.4%)	41 (4.0%)
Training	Strongly Agree	456 (33.5%)	646 (64.3%)
	Somewhat Agree	726 (53.3%)	321 (31.9%)
	Somewhat Disagree	149 (10.9%)	32 (3.2%)
	Strongly Disagree	32 (2.4%)	6 (0.6%)
Frequent nail trims	Strongly Agree	364 (28.5%)	451 (47.3%)
	Somewhat Agree	630 (49.3%)	382 (40.1%)
	Somewhat Disagree	221 (17.3%)	90 (9.4%)
	Strongly Disagree	63 (4.9%)	30 (3.2%)
Nail caps	Strongly Agree	197 (25%)	217 (32.4%)
	Somewhat Agree	327 (41.5%)	267 (39.9%)
	Somewhat Disagree	118 (15%)	99 (14.8%)
	Strongly Disagree	146 (18.5%)	86 (12.9%)
**Attitudes on Scratching Behavior**			
Bothered by scratching	Strongly Agree	113 (8.1%)	4 (0.4%)
	Somewhat Agree	480 (34.2)	30 (2.9%)
	Somewhat Disagree	344 (24.5%)	79 (7.6%)
	Strongly Disagree	454 (32.4%)	916 (88.1%)
	Do not Know	12 (0.9%)	11 (1.1%)
Surrender considered	Strongly Agree	6 (0.4%)	2 (0.2%)
	Somewhat Agree	14 (1%)	1 (0.1%)
	Somewhat Disagree	22 (1.6%)	4 (0.4%)
	Strongly Disagree	1346 (96.5%)	1024 (98.8%)
	Do not Know	7 (0.5%)	5 (0.5%)
Euthanasia considered	Strongly Agree	2 (0.1%)	1 (0.1%)
	Somewhat Agree	1 (0.1%)	0 (0%)
	Somewhat Disagree	3 (0.2%)	4 (0.4%)
	Strongly Disagree	1383 (99.2%)	1026 (99.1%)
	Do not Know	5 (0.4%)	4 (0.4%)
Declaw considered	Strongly Agree	43 (3.1%)	9 (0.9%)
	Somewhat Agree	45 (3.2%)	10 (1%)
	Somewhat Disagree	18 (1.3%)	2 (0.2%)
	Strongly Disagree	1285 (92.1%)	1006 (97.5%)
	Do not Know	4 (0.3%)	5 (0.5%)

**Table 2 animals-12-02551-t002:** Summary statistics comparing enrichment provided by cat owners with cats who show either unwanted scratching behavior (*n* = 1406) or no unwanted scratching behavior (*n* = 1041).

Variable	Levels	Unwanted Scratching	No Unwanted Scratching
**Provision of Scratching Materials/Objects**			
Cardboard	Yes	926 (67.8%)	651 (64.7%)
	No	439 (32.2%)	355 (35.3%)
Sisal rope	Yes	807 (59.1%)	668 (66.4%)
	No	558 (40.9%)	338 (33.6%)
Carpet	Yes	961 (70.4%)	679 (67.5%)
	No	404 (29.6%)	327 (32.5%)
Fabric	Yes	356 (26%)	170 (16.9%)
	No	1009 (73.9%)	836 (83.1%)
Wood	Yes	330 (24.2%)	291 (28.9%)
	No	1035 (75.8%)	715 (71%)
Scratching post	Yes	960 (70.4%)	716 (71.3%)
	No	404 (29.6%)	288 (28.7%)
Cat tree	Yes	857 (62.8%)	663 (66%)
	No	507 (37.2%)	341 (34%)
Flat surface	Yes	827 (60.6%)	608 (60.6%)
	No	537 (39.4%)	396 (39.4%)
Hanging scratching surface	Yes	248 (18.1%)	154 (15.3%)
	No	1116 (81.8%)	850 (84.7%)
**Other Enrichment**			
Active play time	0–1 h	1136 (80.8%)	804 (77.3%)
	1–2 h	225 (16%)	174 (16.7%)
	>2–3 h	23 (1.63%)	36 (3.46%)
	>3 h	22 (1.6%)	26 (2.5%)
Active interaction time	0–1 h	338 (24%)	250 (24%)
	1–2 h	390 (27.8%)	272 (26.2%)
	>2–3 h	259 (18.4%)	175 (16.8%)
	>3–4 h	138 (9.82%)	118 (11.3%)
	>4–5 h	86 (6.12%)	77 (7.4%)
	>5 h	194 (13.8%)	148 (14.2%)
Outdoor access	Yes	384 (27.6%)	339 (33%)
	No	1006 (72.6%)	689 (67%)
Controlled outdoor access	Yes	261 (58%)	204 (52%)
	No	189 (42%)	188 (48%)
Uncontrolled outdoor access	Yes	165 (36.7%)	165 (42.1%)
	No	285 (63.3%)	227 (57.9%)

**Table 3 animals-12-02551-t003:** Logistic regression models of cat characteristics (*n* = 2446), owner demographic factors and management (*n* = 2434), and provisions of enrichment (*n* = 2218) associated with unwanted scratching behavior.

Variables		OR ^a^	95% CI ^b^	*p*-Value
**Model 1: Provisions of Enrichment**				
Use fabric material	Yes vs. No	3.15	2.55, 3.90	<0.0001
Use flat surfaces	Yes vs. No	0.83	0.7, 0.99	0.037
Outdoor access	Outdoor vs. Indoor	0.78	0.64, 0.94	0.01
Use sisal rope	Yes vs. No	0.73	0.61, 0.86	<0.0001
**Model 2: Owner Characteristics and Management**				
Provide additional scratching posts to prevent unwanted scratching	Yes vs. No	0.62	0.50, 0.76	<0.0001
Restrict access to areas where unwanted scratching occurs	Yes vs. No	0.6	0.48, 0.74	<0.0001
Owner deters unwanted scratching using verbal correction	Yes vs. No	1.56	1.27, 1.92	<0.0001
Owner deters unwanted scratching using physical correction	Yes vs. No	1.29	1.07, 1.56	0.007
Owner interrupts unwanted scratching	Yes vs. No	1.48	1.19, 1.85	0.001
Owner rewards use of appropriate scratching objects	Yes vs. No	0.78	0.65, 0.94	0.007
Owner applies attractant to preferred scratching areas	Yes vs. No	0.68	0.57, 0.81	<0.0001
**Model 3: Cat Characteristics**				
Age				<0.00001
	<4–12 months vs. 1–6 years	1.57	1.00, 2.48	0.052
	7–10 years vs. 1–6 years	0.78	0.64, 0.95	0.015
	11–14 years vs. 1–6 years	0.65	0.51, 0.82	<0.0001
	15+ years vs. 1–6 years	0.48	0.34, 0.68	<0.0001

^a^ Odds ratio based on the output of logistic regression models; ^b^ 95% confidence interval of the odds ratio.

## Data Availability

Due to ethical restrictions, data is only available by request from the corresponding author.

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
