# Peer review of "Unwanted Scratching Behavior in Cats: Influence of Management Strategies and Cat and Owner Characteristics"

_animals, 2022, doi:10.3390/ani12192551_

Round 1
Reviewer 1 Report
The simple summary implies that the owners influence cat scratching, but mostly the survey dealt with responses to unwanted scratching and techniques to lower the possibility of unwanted scratching
cat trees should be mentioned
34 flat scratching surfaces
79 Do you mean Feliscratch ( now off the market in US) or Feliway)89 scratch grass?
Table 2 is hard to read although Table 1 is OK can it be reformatted so a ruler isn't necessary to determine variable level and unwanted scratching for any particular variable
205 presumably a scratching surface cardboard or sisal on flat surface.
227 and 270 fabric ? Do you mean cotton fabric versus carpeting or carpeting vs sisal. or do you mean cats were scratching fabric such as curtains as a type of unwanted scratching?
286 isn't this circular? if the cats didn't scratch the owner wouldn't punish them. I applaud the finding of facts that will deter owners from punishing their cats, but there may be a problem with cause and effect.
309 don't have an influence on increasing or decreasing the unwanted scratching
319 why would declawed not be in your sample; might reiterate that owners of declawed cats were not part of survey.
In discussion might mention that cats tend to scratch in prominent places in a room presumably as territorial statements.
Author Response
Unwanted Scratching Behavior in Cats: Influence of Management Strategies and Cat and Owner Characteristics
Thank you for reviewing the corresponding manuscript, and for your helpful comments and suggestions. In light of these comments, we have made a number of improvements in the manuscript. Please see below for responses to each reviewer comment.
Comments and Suggestions for Authors
The simple summary implies that the owners influence cat scratching, but mostly the survey dealt with responses to unwanted scratching and techniques to lower the possibility of unwanted scratching.
- cat trees should be mentioned
See L32 for revision.
Revised text (L31-32):
Owner perspectives and management strategies aligned with current recommendations as they preferred to use appropriate surfaces (e.g., cat trees)…
- 34 flat scratching surfaces
See L35 for correction.
- 79 Do you mean Feliscratch (now off the market in US) or Feliway)
See L87-89 for clarification.
Revised text (L87-89):
The application of feline pheromones (specifically, feline interdigital semiochemical marketed as Feliscratch by Feliway) has been found effective in…
- 89 scratch grass?
See L98 for correction; grass was substituted for ‘posts’.
Revised text (L98):
…with its additional opportunities and items available for scratching, such as trees, posts, and fences.
- Table 2 is hard to read although Table 1 is OK. Can it be reformatted so a ruler isn't necessary to determine variable level and unwanted scratching for any particular variable?
See Table 2 for a revised version, with additional spacing between variables and shorter distances between data and the associated variables.
- 205 presumably a scratching surface cardboard or sisal on flat surface.
See L224 for clarification. Flat scratching surfaces, made of any material (e.g., cardboard, sisal, etc.) that is given to cats to promote scratching on the ground. This contrasts with vertical scratching surfaces like scratching posts.
- 227 and 270 fabric ? Do you mean cotton fabric versus carpeting or carpeting vs sisal. or do you mean cats were scratching fabric such as curtains as a type of unwanted scratching?
In the survey, fabric is referred to in more broad terms. The idea was to identify which materials cats are more likely to use, and how the use of these specific materials is associated with unwanted scratching. As such, fabric itself is broadly referencing any fabric the cat is either provided by the owner or has access to use to scratch. We have added clarification to L296.
Revised text (L296):
If owners reported observing their cat scratching fabric materials (e.g., cotton), owners …
- 286 isn't this circular? if the cats didn't scratch the owner wouldn't punish them. I applaud the finding of facts that will deter owners from punishing their cats, but there may be a problem with cause and effect.
Thank you for this helpful comment. We agree this is circular and not portraying all the possible ways these two variables may be related. We have provided additional text detailing the possible directions of the relationship to help explain the result.
Revised text (L381-385):
Though results may suggest that positive punishment may increase unwanted scratching, it could also be that cats that perform unwanted scratching are more likely to receive physical or verbal correction from their owner; further research is needed to determine the direction of this relationship.
- 309 don't have an influence on increasing or decreasing the unwanted scratching
See revision on L398-399.
Revised text (L398-399):
…suggesting these strategies do not have an influence on increasing or decreasing owner reports of unwanted scratching behavior.
- 319 why would declawed not be in your sample; might reiterate that owners of declawed cats were not part of survey.
Thank you for this suggestion; see revision on L408.
Revised text (L408):
Additionally, this finding may be a result of survival bias as older cats may have been removed from our sample as a result of declaw (as declawed cats were removed from our sample), relinquishment, or euthanasia.
- In discussion might mention that cats tend to scratch in prominent places in a room presumably as territorial statements.
Thank you for this suggestion, we have expanded the description of the use of scratching for territory marking in the introduction on L61-62 as well as included information regarding the need for further research on the relationship between scratching materials and the purpose of scratching behaviors, such as territory marking on L357-360.
Revised text:
L61-62:
…marking territory for communication with other cats through pheromones found in the interdigital glands, sharpening claws, and removing…
L357-360:
Cats use scratching for a variety of reasons, including communicating with other cats for territory marking [8], thus further research may explore whether the type and location of scratching material influences their scratching behavior.

Reviewer 2 Report
Since cats living in Northern American countries, have been massively declawed, for so many years, the topic of the paper is of major interest. Unfortunately, the structure of the questionnaire, and its statistical analysis, fail in giving really new and interesting information, despite the very broad population enrolled. The initial purpose of this paper, was clearly to identify "risk profiles", to facilitate the work of the vets and behavior councellors, supposed to assist cat's owners. It would have been much more interesting to use multivariate analysis, in the attempt of detecting profiles. The current results do not really provide new information for behavior-medicine.
I recommend to try to move to multivariate analysis.
Author Response
Unwanted Scratching Behavior in Cats: Influence of Management Strategies and Cat and Owner Characteristics
Thank you for reviewing the corresponding manuscript, and for your helpful comments and suggestions. In light of these comments, we have made a number of improvements in the manuscript. Please see below for responses to each reviewer comment.
Comments and Suggestions for Authors
Since cats living in Northern American countries, have been massively declawed, for so many years, the topic of the paper is of major interest. Unfortunately, the structure of the questionnaire, and its statistical analysis, fail in giving really new and interesting information, despite the very broad population enrolled. The initial purpose of this paper, was clearly to identify "risk profiles", to facilitate the work of the vets and behavior counselors, supposed to assist cat's owners. It would have been much more interesting to use multivariate analysis, in the attempt of detecting profiles. The current results do not really provide new information for behavior-medicine.
I recommend to try to move to multivariate analysis.
Thank you for taking the time to read our manuscript and for providing this comment and suggestion. When referring to multivariate analysis to capture risk profiles, we are assuming that you are referring to combining the variables into one single model. In light of this suggestion, we performed the analysis again combining all variables into one single model, and the results did not change from the current results presented in the submitted manuscript. If we are incorrect as to what you are suggesting, please let us know any specific details regarding the type of analysis that is being recommended.
The authors initially started with this approach; however, to avoid a large model (testing 35+ variables) and thus avoid the increased chances of errors associated with large models (type I error), we decided to make the models more focused on specific questions related to cat scratching. As the goal of this research was to identify which specific factors related to the cat, the owner, and their management styles influence cat scratching behavior, this method allowed us to properly evaluate these factors without having to reduce the number of variables tested and without compromising model quality.
In addition to identifying owner perceptions on how to manage this commonly reported cat behavioral issue, our analysis provides the first empirical evidence in support of the following: deterring owners from the use of positive punishment training in response to unwanted cat scratching, encouraging owners to redirect cat scratching behavior towards appropriate objects and materials, the use of positive reinforcement training, and providing enrichment items for reducing unwanted cat scratching. To the best of our knowledge, this information has not been previously published and therefore adds to the current literature surrounding cat behavioral issues; this is new information can be used to help support current recommendations in responding to unwanted cat behavior.

Reviewer 3 Report
This article describes results of a survey of ~2400 cat owners in the US and Canada. They described their cat's scratching behaviour and regressions were used to calculate factors influencing this behaviour. This is an interesting people which should be of interest to the journal's readership. Before I can recommend it for publication, I have a few suggestions.
Major comments
The authors found that participants were generally unlikely to consider declawing as an option for scratching management. However, they excluded 361 participants who had already declawed their cat. I appreciate that this exclusion was because the dependent variable was causing damage due to scratching. So it doesn't make sense to include participants whose cats lack the capacity to cause damage. Nonetheless, this could have led to a sampling error, especially related to that question. This is a serious limitation that needs to be acknowledged, and perhaps the more permanent management solutions (ie, declaw, surrender, euthanise) should just be excluded from analysis/reporting due to this sampling error. At a minimum, declawing should be excluded.
The data collection segment of the Methods does not provide sufficient information about the recruitment process. What did the recruitment ad say? What sorts of groups were targeted on social media? In the Discussion, the authors mention animal welfare groups, but this is not described anywhere in the methods.
Were reliability analyses done on the items that were combined into composite variables? If so, these results should be reported. If not, this needs to be justified clearly. A reliability analysis would be far preferable here.
Other comments
in the abstract, simple summary, and intro, the authors describe scratching as 'highly motivated'. What is meant by this? Do they mean to say 'highly self-motivated'? or 'internally motivated'? Or 'inherently self-rewarding'? I believe the authors mean that this behaviour is something that cats do just because they like it, as opposed to something that they are just highly motivated to do (e.g., due to external motivations). Sometimes people are highly motivated to go to work because they'll get paid, even if they don't particularly like the work they do. So 'highly motivated' could be worded more accurately.
Simple summary - L10 'natural behaviours' should be 'natural behaviour' without the 's' at the end.
Abstract - L36 do the authors mean to write p < 0.007? Right now it is 0.07 which would not be significant.
Keywords - the keywords do not need to be the same as words in the title, because words in the title are automatically indexed in search engines. There's a good website to provide advice on this: https://getproofed.com.au/writing-tips/how-to-pick-the-best-keywords-for-a-journal-article/ It's becoming increasingly important to get these right because systematic reviews are becoming more and more common.
Intro
L55 - please briefly describe what is meant by 'removing sheaths'
L84-86 - sentence fragment.
Results
The order of the response options is strange. It goes from strongly agree - somewhat agree - strongly disagree - somewhat disagree. Aren't the last two in the wrong order? Was this the order in the survey itself? If so, it might be a problem if people are expecting a scale and the order is wrong. Please clarify. Also, suggest changing the order in the table so that they are in the right order.
Discussion
L257 - but see my comment above re: declawing and sampling bias.
Conclusion
L346 and 347 - please change the phrasing from causal (e.g., 'may lead to', 'may reduce') to correlational (e.g., 'is associated with').
Author Response
Unwanted Scratching Behavior in Cats: Influence of Management Strategies and Cat and Owner Characteristics
Thank you for reviewing the corresponding manuscript, and for your helpful comments and suggestions. In light of these comments, we have made a number of improvements in the manuscript. Please see below for responses to each reviewer comment.
Comments and Suggestions for Authors
This article describes results of a survey of ~2400 cat owners in the US and Canada. They described their cat's scratching behaviour and regressions were used to calculate factors influencing this behaviour. This is an interesting people which should be of interest to the journal's readership. Before I can recommend it for publication, I have a few suggestions.
Major comments
The authors found that participants were generally unlikely to consider declawing as an option for scratching management. However, they excluded 361 participants who had already declawed their cat. I appreciate that this exclusion was because the dependent variable was causing damage due to scratching. So it doesn't make sense to include participants whose cats lack the capacity to cause damage. Nonetheless, this could have led to a sampling error, especially related to that question. This is a serious limitation that needs to be acknowledged, and perhaps the more permanent management solutions (ie, declaw, surrender, euthanise) should just be excluded from analysis/reporting due to this sampling error. At a minimum, declawing should be excluded.
Thank you for this comment. Though cat owners in the sample did not previously choose to declaw their cat or do not have a declawed cat, it is still possible that they would consider declawing in the future. In the survey, owners were asked to rate their agreement (using a scale of strongly agree to strongly disagree) with the following statement, “I have considered declawing my cat due to its scratching behaviour.” As such, we believe it is important to report the results from this response, as declawing is predominately preformed to either prevent or resolve scratching issues. By incorporating this data, we can fully report owner perceptions on managing scratching issues, from providing enrichment (e.g., scratching posts) to considering more permanent solutions (e.g., declaw).
The data collection segment of the Methods does not provide sufficient information about the recruitment process. What did the recruitment ad say? What sorts of groups were targeted on social media? In the Discussion, the authors mention animal welfare groups, but this is not described anywhere in the methods.
Thank you for noticing this gap in our Methods. The recruitment ad stated that we are currently recruiting cat owners to participate in a survey regarding scratching. It specified the inclusion criteria and how long the survey would take and provided a link to the survey. No specific groups were targeted on social media; this word choice has been corrected in the manuscript. During recruitment, we tried to broaden our outreach and began to contact other animal behavior and welfare labs asking them to share our advertisement on their social media pages, and we advertised our survey on various groups on social media that were cat-related or operated by cat owners. We have provided clarification as to our recruitment process within the methods.
Revised text:
(L119-123):
Virtual snowball sampling was used to distribute the survey, as the initial advertisement (which called owners to participate in a survey regarding scratching) was shared via Facebook and sharing was encouraged. To encourage outreach, the advertisement was shared to other behavior and welfare and cat-related social media pages.
(L415-417)
Further, the distribution of the survey was conducted through social media that reached other cat-related social media pages and other animal welfare organizations.
Were reliability analyses done on the items that were combined into composite variables? If so, these results should be reported. If not, this needs to be justified clearly. A reliability analysis would be far preferable here.
We appreciate this suggestion. Reliability analysis was not believed necessary to perform as the variables were all linked to one specific question; the response options to one question were the variables that were combined into the composite variable. The question asked owners if they have had volunteer or paid positions related to cats. To answer this question, they were asked to select all that apply to the following list of experiences: cat sitter, cat breeder, cat trainer/behaviour consultant, veterinarian, veterinary technician, cat boarding facility worker, shelter worker, groomer, researcher, and foster parent for a rescue organization. As we knew that having cat experience of any sort could influence their survey responses, but we were not directly interested in detecting the differences between each type of experience, we reduced the variable to a yes or no option, whether or not they had cat-related experience. We have added comments addressing this in L155-157, and in limitations L429-433.
Revised text:
(L155-157):
As these variables were all directly linked to one survey question regarding cat-related experience, these variables were highly conceptually related and thus a reliability analysis was not deemed necessary.
(L429-433):
Also, by combining the different levels of cat experience into a single variable, ‘experience’, we were unable to confirm whether different levels of experience influence different perceptions and different management strategies employed by cat owners; for future research a more detailed analysis would be needed to determine this.
Other comments
in the abstract, simple summary, and intro, the authors describe scratching as 'highly motivated'. What is meant by this? Do they mean to say 'highly self-motivated'? or 'internally motivated'? Or 'inherently self-rewarding'? I believe the authors mean that this behaviour is something that cats do just because they like it, as opposed to something that they are just highly motivated to do (e.g., due to external motivations). Sometimes people are highly motivated to go to work because they'll get paid, even if they don't particularly like the work they do. So 'highly motivated' could be worded more accurately.
Thank you for this suggestion; we have provided clarification on L10 and L22.
Revised text:
(L10):
Cat scratching is a self-motivated and natural behavior …
L22:
Despite scratching behavior in owned domestic cats being a self-motivated and natural behavior, …
Simple summary - L10 'natural behaviours' should be 'natural behaviour' without the 's' at the end.
See L10 for the correction.
Abstract - L36 do the authors mean to write p < 0.007? Right now it is 0.07 which would not be significant.
Thank you for this observation, the p-value was corrected on L36.
Keywords - the keywords do not need to be the same as words in the title, because words in the title are automatically indexed in search engines. There's a good website to provide advice on this: https://getproofed.com.au/writing-tips/how-to-pick-the-best-keywords-for-a-journal-article/ It's becoming increasingly important to get these right because systematic reviews are becoming more and more common.
We appreciate this comment and the useful link. The keywords have been updated on L43.
Revised text (L43):
Keywords: cat; scratching; enrichment; training; welfare; behavior
Intro
L55 - please briefly describe what is meant by 'removing sheaths'
See L62-63 for clarification.
Revised text (L62-63):
… sharpening claws, and removing claw sheaths (the outer layer of the claw).
L84-86 - sentence fragment.
See L93-94 for correction.
Revised text (L93-94):
It is believed that inappropriate scratching has become more problematic within households as a result of restricting cats indoors.
Results
The order of the response options is strange. It goes from strongly agree - somewhat agree - strongly disagree - somewhat disagree. Aren't the last two in the wrong order? Was this the order in the survey itself? If so, it might be a problem if people are expecting a scale and the order is wrong. Please clarify. Also, suggest changing the order in the table so that they are in the right order.
The order in the survey was Strongly Agree, Somewhat Agree, Somewhat Disagree, Strongly Disagree. Table 2 has been corrected to reflect this order.
Discussion
L257 - but see my comment above re: declawing and sampling bias.
Thank you for this comment. Though cat owners in the sample did not previously choose to declaw their cat or do not have a declawed cat, it is still possible that they would consider declawing in the future. In the survey, owners were asked to rate their agreement (using a scale of strongly agree to strongly disagree) with the following statement, “I have considered declawing my cat due to its scratching behaviour.” As such, we believe it is important to still report the results from this response, as declawing is predominately preformed to either prevent or resolve scratching issues. By incorporating this data, we can fully report owner perceptions on managing scratching issues, from providing enrichment (e.g., scratching posts) to considering more permanent solutions (e.g., declawing).
Conclusion
L346 and 347 - please change the phrasing from causal (e.g., 'may lead to', 'may reduce') to correlational (e.g., 'is associated with').
See L447-449 for correction.
Revised text (L447-449):
… for instance, the use of positive punishment techniques was associated with more reports of unwanted scratching while providing enrichment and positive reinforcement techniques was associated with fewer reports of unwanted scratching.

Round 2
Reviewer 2 Report
Accept in its current form
Author Response
Thank you for taking the time to review and provide helpful suggestions for improving this manuscript.
Reviewer 3 Report
this ms has been much improved. I just have one further recommendation before publication.
1. I take the authors' point about leaving the question in about declawing, even though they excluded people who had already declawed their cats. However, this should be noted in the Discussion as a limitation of the study. There is still the possibility of a sampling error due to this exclusion.
Otherwise, nice work. It's a good study.
Author Response
Thank you for this suggestion; we have added this to our limitations section.
Revised text (L429-431):
Sampling error could also have arisen from the exclusion of participants who reported declawing their cat, though this was deemed necessary as these participants would not be able to report damage to scratching.